 SHORT REPORT

# Matrix-trapped viruses can prevent invasion of bacterial biofilms by colonizing cells

**Matthew C Bond[1], Lucia Vidakovic[2], Praveen K Singh[2], Knut Drescher[2,3,4], Carey D Nadell[1]***

[1]Department of Biological Sciences, Dartmouth College, Hanover, United States; [2]Max Planck Institute for Terrestrial Microbiology, Marburg, Germany; [3]Department of Physics, Philipps University Marburg, Marburg, Germany; [4]Biozentrum, University of Basel, Basel, Switzerland

**Abstract** Bacteriophages can be trapped in the matrix of bacterial biofilms, such that the cells inside them are protected. It is not known whether these phages are still infectious and whether they pose a threat to newly arriving bacteria. Here, we address these questions using *Escherichia coli* and its lytic phage T7. Prior work has demonstrated that T7 phages are bound in the outermost curli polymer layers of the *E. coli* biofilm matrix. We show that these phages do remain viable and can kill colonizing cells that are T7-susceptible. If cells colonize a resident biofilm before phages do, we find that they can still be killed by phage exposure if it occurs soon thereafter. However, if colonizing cells are present on the biofilm long enough before phage exposure, they gain phage protection via envelopment within curli-producing clusters of the resident biofilm cells.

***For correspondence:**
carey.d.nadell@dartmouth.edu

**Competing interests:** The authors declare that no competing interests exist.

## Introduction

Bacteria and their bacteriophage predators, or phages, are found alongside each other in nearly every environment examined (*Rodriguez-Brito et al., 2010*; *Zablocki et al., 2016*; *Clokie et al., 2011*; *Manrique et al., 2016*; *Correa et al., 2021*). Phages inject their genomes into the cytoplasm of their hosts, and in the case of obligate lytic phages, immediately begin co-opting host resources to replicate. Eventually, host cells are lysed to release a new cohort of phage virions. Predatory pressure from phage attack drives bacterial evolution, diversification, and ultimately the community structure of many microbiomes (*Blazanin and Turner, 2021*; *Harrison et al., 2013*; *Lenski and Levin, 1985*; *Abedon, 2008*; *Koskella and Brockhurst, 2014*; *Gómez and Buckling, 2011*; *Keen and Dantas, 2018*). The mechanistic, ecological, and evolutionary features of phage-bacteria interactions have a deep history of study, including many seminal theoretical and experimental papers that have characterized the population and evolutionary dynamics of phage-bacteria interactions (*Koskella and Brockhurst, 2014*; *Abedon, 2009*; *Chao et al., 1977*; *Susskind and Botstein, 1978*). The traditional literature in this area mostly considers well-mixed culture conditions such as those in shaken liquids, which can reveal fundamental aspects of phage-bacteria interactions without spatial structure. However, in nature bacteria often reside in spatially constrained, surface-bound communities, or biofilms (*Nadell et al., 2016*; *Hellweger et al., 2016*; *Flemming and Wingender, 2010*; *Flemming et al., 2016*). The limited work that has focused on phage infection in biofilm environments has often found outcomes that differ substantially from those observed in mixed liquid environments (*Hansen et al., 2019*; *Levin and Bull, 2004*; *Simmons et al., 2020*; *Davies et al., 2016*; *Abedon, 2016*; *Schrag and Mittler, 1996*; *Eriksen et al., 2018*; *Simmons et al., 2018*; *Chaudhry et al., 2018*; *Pires et al., 2021*; *Vidakovic et al., 2018*).

A defining feature of biofilm populations is the presence of a self-secreted adhesive polymer matrix that modulates cell-cell and cell-surface interactions, in addition to influencing collective biofilm architecture (*Nadell et al., 2016*; *Flemming and Wingender, 2010*; *Erskine et al., 2018*; *Teschler et al., 2015*; *Hartmann et al., 2019*; *Colvin et al., 2012*). Several recent papers have demonstrated that the biofilm matrix can be central to phage-host coevolution. *Pseudomonas fluorescens* and *Escherichia coli* rapidly evolve mucoid colony phenotypes – which reflect increased and/or altered matrix secretion – when they are under phage attack (*Scanlan and Buckling, 2012*; *Chaudhry et al., 2020*). Curli fibers, a proteinaceous component of the *E. coli* matrix, can block biofilm-dwelling cells from T7 and T5 phage exposure (*Vidakovic et al., 2018*). In this case, phages can be seen directly enmeshed in the curli mesh without infecting biofilm cells unless the integrity of the curli layer is compromised (*Simmons et al., 2020*; *Vidakovic et al., 2018*). Recent papers by *Darch et al., 2017*, *Díaz-Pascual et al., 2019*, and *Dunsing et al., 2019*, respectively, suggest a similar pattern of matrix-dependent phage protection in *Pseudomonas aeruginosa*, *Vibrio cholerae*, and *Pantoea stewartii*. Earlier work from *Kay et al., 2011* used plaque-forming unit (PFU) count assays to show that if phage-exposed biofilms are disassociated by a matrix-degrading agent, the phage titer in the surrounding media increases, suggesting that phages released from the matrix are potentially still infectious. It is not known, however, whether phages remain threatening to newly arriving bacteria while the phages are still embedded in the intact biofilm matrix; it could be, for example, that matrix-embedded phages are mostly degraded or trapped in configurations that render them unable to infect host cells. The answer to this question may have a significant impact on the processes of population assembly as cells encounter and attempt to colonize pre-existing biofilms.

Inspired by the findings above, we investigated the consequences of phage entrapment in the matrix for biofilm population assembly. If they remain infectious, these phages could pose a threat to new cells that attempt to colonize the biofilm surface. Here, we explored this possibility by studying how matrix-embedded phages influence the invasion of bacteria into pre-existing biofilm populations, and whether biofilm-invading cells can integrate into the existing matrix and gain phage protection. To address these questions, we used a combination of microfluidic culture, phage infection reporter techniques, high resolution confocal microscopy, and detailed spatial analysis of the resulting image data. We find that matrix-trapped phages do indeed remain infectious and reduce the ability of newly arriving cells to colonize existing biofilms, and that this effect is dependent on the relative timing of the arrival of phages and colonizing cells.

## Results and discussion

### Influence of matrix-trapped phages on invading planktonic cells

*E. coli* produces a variety of matrix components at different times during biofilm formation, including flagellar filaments, curli amyloid fibers, and polysaccharides such as colanic acid and cellulose (*Vidakovic et al., 2018*; *Beloin et al., 2008*; *Barnhart and Chapman, 2006*; *Evans and Chapman, 2014*; *Hammar et al., 1996*; *Pesavento et al., 2008*; *Serra et al., 2013a*; *Serra et al., 2013b*). Curli fibers are the single matrix component known to be essential for T7 phage protection in *E. coli* biofilms (*Vidakovic et al., 2018*) and are assembled by extracellular polymerization of CsgA monomers on an outer membrane baseplate comprised of CsgB (*Evans and Chapman, 2014*; *Barnhart and Chapman, 2006*). Curli proteins secreted by *E. coli* localize primarily in the upper half of the biofilm-dwelling population, and introduced phages become trapped in the outer part of this curli matrix layer (*Vidakovic et al., 2018*; *Serra et al., 2013a*; *Serra et al., 2013b*; *Figure 1A*). The biofilm-dwelling cells amongst and below the curli fiber matrix layer thus become protected against T7 exposure in the absence of the evolution of physiological resistance to T7 phages (*Simmons et al., 2020*; *Vidakovic et al., 2018*). Although these phages are blocked from diffusing into the biofilm interior, we hypothesized that trapped phage particles may remain capable of infecting cells that reach them from the liquid environment that surrounds the biofilm. To test this hypothesis, we first grew biofilms of *E. coli* AR3110 for 62 hr (*Figure 1B*; *Figure 1—figure supplement 1*); prior work has established this growth period as optimal for full curli expression along the biofilm front (*Vidakovic et al., 2018*), and we confirmed this result in our microfluidic growth conditions (*Figure 1—figure supplement 2*). The resident cells harbored a chromosomal construct for constitutive expression of the far-red fluorescent protein mKate2 (*Shcherbo et al., 2007*) to make them visible

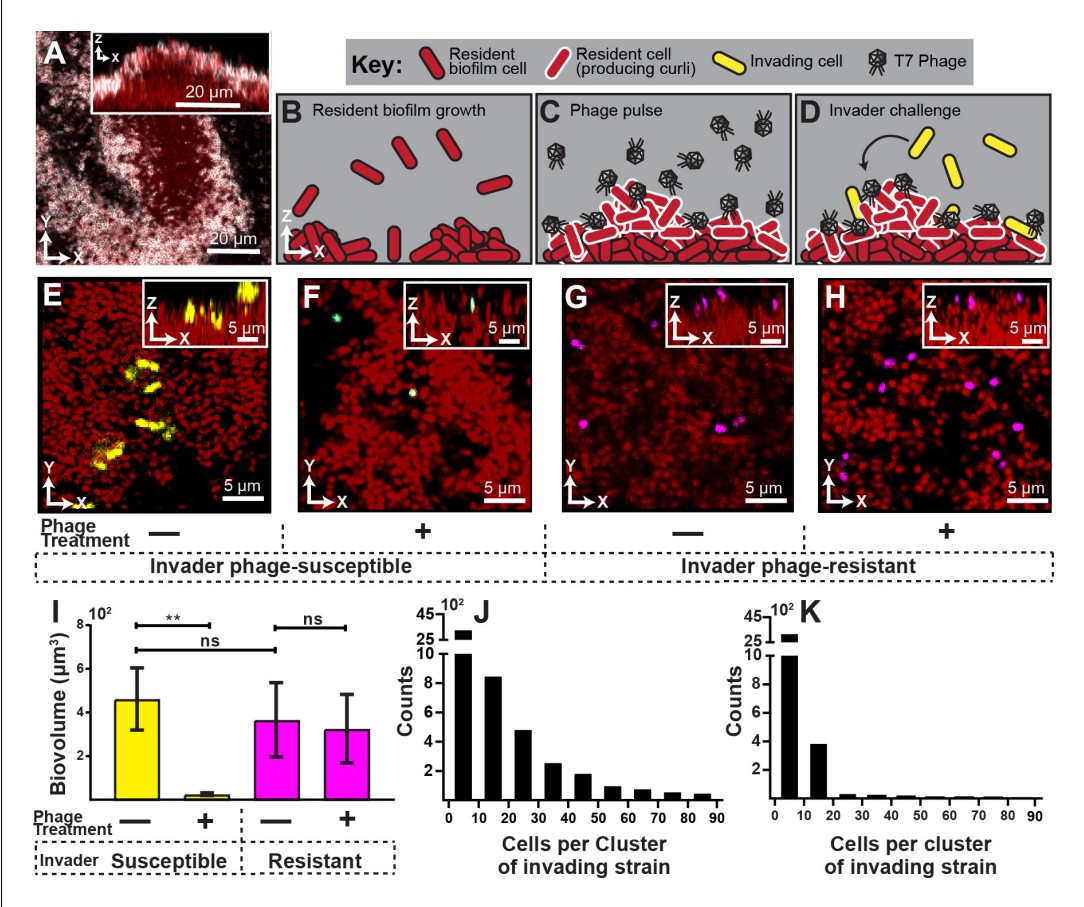

**Figure 1.** Visualization and quantification of biofilm invasion with or without phage exposure. (A) Visualization of *E. coli* biofilm (red) with stained curli matrix (white), including one x-y optical section (main image) and z-projection (inset). (B–D) Illustration of invasion assay procedure. (B) Resident biofilm-producing cells (red) were allowed to grow for 62 hr prior to phage exposure. (C) Inlet tubing was then swapped for 1 hr to new tubing and syringe containing a concentrated phage suspension ($2\times10^8$ PFU/mL). (D) Resident biofilms were then challenged with isogenic *E. coli* expressing a different fluorescent protein (yellow); this was performed by swapping for 3 hr to new tubing and syringe containing high density *E. coli* culture ($OD_{600}$ = 6.0). Biofilms were imaged 10 hr following this step. (E) Invading cells (yellow) can successfully attach to the periphery of resident biofilm (red) in the absence of phages. (F) Invading cells fail to colonize when biofilms are pretreated with phages, which become trapped in the biofilm matrix. The phage-encoded reporter (cyan) indicates invading cells that have become phage-infected. (G,H) An *E. coli* mutant that does not permit phage amplification (Δ*trxA*, magenta) invades equally well in control and phage pre-treatment conditions. (I) Quantification of image data shown in E-H; average invading biovolume per field of view (150 μm x 150 μm x 15 μm; length x width x height). Error bars represent SEM. Pairwise comparisons were performed with Mann-Whitney Signed Ranks tests with Bonferroni correction (n = 4-6 biological replicates; ** denotes $p<0.05$). (J–K) Invading cell cluster size distributions for phage-susceptible cells invading biofilms without (J) or with (K) phage pre-treatment (n = 4-6 biological replicates).

The online version of this article includes the following source data and figure supplement(s) for figure 1:

**Source data 1.**

**Figure supplement 1.** Method for phage pretreatment experiments.

**Figure supplement 2.** Time course imaging of *E. coli* biofilms documenting the course of curli matrix protein accumulation.

**Figure supplement 2—source data 1.**

with fluorescence time-lapse microscopy. The biofilms were then exposed to a 1 hr pulse of media containing lytic T7 phages at $2\times10^8$ PFU/mL, such that phages could accumulate in the outer matrix curli layer (*Figure 1C*). The phages contained a *sfGFP* (*Pédelacq et al., 2006*) expression construct engineered into their genome, allowing us to visualize any cells that became phage-infected (*Vidakovic et al., 2018*). For a control comparison, we performed the same procedure but exposed the resident biofilms to sterile media containing no phages. Following phage exposure or control treatment, we performed a population invasion step in which high-density planktonic cultures of phage-susceptible wild type *E. coli* AR3110 ($OD_{600}$ = 6.0) were added to the chambers for 3 hr to

colonize the resident biofilms (*Figure 1D*). The full experimental procedure for these experiments is summarized in *Figure 1—figure supplement 1*. Colonizing cells were isogenic to resident wild-type cells, except they constitutively expressed mKO-κ (*Tsutsui et al., 2008*) so that they could be distinguished from resident cells during imaging.

We found that in the absence of phage exposure, resident biofilms could be colonized by planktonic cells, albeit not at high efficiency (*Figure 1E,I*). The colonizing cells were restricted to the outer surface of the resident biofilms and could not enter the biofilm interior, similar to what we have seen previously for *V. cholerae* (*Nadell et al., 2015*). However, when resident biofilms were pre-exposed to phages, colonization by phage-susceptible cells was almost completely eliminated (*Figure 1F,I*). Further, most invading cells that were detected on phage-exposed biofilms were fluorescent in the sfGFP channel, indicating that they had been phage-infected but not yet lysed.

Our interpretation of this result is that susceptible invading cells encounter phages in the curli matrix, become infected, and lyse to release new phages, or fail to divide further. We assessed this idea quantitatively by measuring the cell cluster size distributions of invading cells on biofilms with or without phage-pre-exposure. Without pre-exposure, we found numerous instances of cell clusters with 20 cells or more, indicating several rounds of division in the 10 hr between invasion and imaging (*Figure 1J*). On biofilms exposed to phages prior to invasion of new *E. coli* cells, there were almost no groups larger than one or a few cells (*Figure 1K*), indicating few or no divisions in the period between invasion and imaging. This is consistent with the observation noted above that on phage pre-exposed biofilms, any remaining invading cells were expressing the phage infection reporter, which would preclude further growth and division.

Another (though not mutually exclusive) explanation for our results is that phages, by occupying potential sites of attachment, block the physical interaction of invading cells with the biofilm outer surface. We tested this possibility by repeating the experiment above with invading cultures of an *E. coli* mutant harboring a clean deletion of *trxA;* this strain does not allow for phage amplification (*Qimron et al., 2006*). The Δ*trxA* deletion mutant was found to invade resident biofilms at equal rates whether or not the resident was pre-exposed to T7 phages, which suggests that phages do not block attachment sites for invading cells (*Figure 1G,H,I*). Noting that the Δ*trxA* deletion mutant undergoes abortive infection upon exposure to phage T7, the fact that Δ*trxA* can colonize resident biofilms, while WT *E. coli* cannot, indicates that phage amplification from initial sites of infection may be important for elimination of phage-susceptible invading cells.

Phages that are trapped in the curli matrix of *E. coli* biofilms are thus persistent as a threat to incoming susceptible bacteria. This result also implies that in the event that curli-protected cells within resident biofilms disperse individually or *en masse*, they may too be susceptible to phages that are released from the biofilm exterior. This remains an important question for future work.

## Invading cells gain phage protection, after a delay

Once we determined that matrix-embedded phages could infect recently attached susceptible cells, we asked whether invading cells fare better if they arrive prior to the phage pulse. To explore this question, we again grew resident biofilms of *E. coli* AR3110 for 62 hr, followed by a planktonic population invasion step as described above. Chambers were separated into two groups: In the first group, phages were pulsed into the chamber immediately after colonization by the invading cell population. In the second group, biofilms were incubated post-invasion for 10 hr – the time at which we observed the colonized biofilms to reach a population steady-state – and then subjected to a phage pulse. This experimental procedure is summarized in *Figure 2—figure supplement 1*. Images were taken of each chamber approximately 10 hr after the phage pulse, allowing for multiple infection and lysis cycles to occur and the system to reach its new equilibrium. When phage pulses occurred immediately after biofilms were colonized by an invading strain, the invading cells were mostly killed by T7 phage exposure (*Figure 2A,B*). Interestingly, however, invading cells were not significantly affected by phage pulses that arrived 10 hr after the colonizing strain attached to the resident biofilm outer periphery (*Figure 2A,C*).

## Invading cells indirectly co-opt matrix of resident biofilms

Having observed that invading cells gain phage protection, but only after a delay between attaching to a resident biofilm and phage exposure, we next investigated how invading cells gain phage

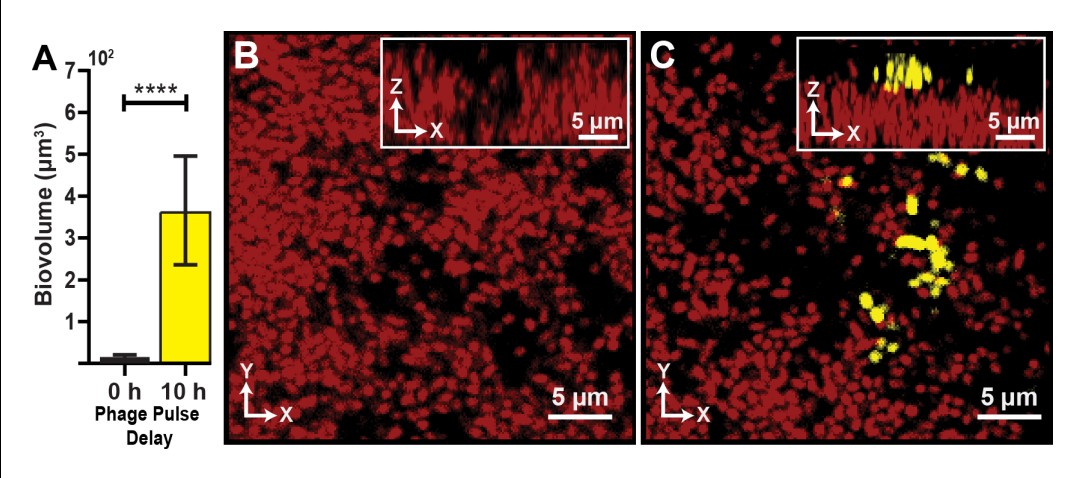

**Figure 2.** Visualization and quantification of colonization success with phage exposure post-colonization. (**A**) Average invading biovolume per field of view (150 µm x 150 µm x 15 µm; length x width x height). Error bars represent SEM. Pairwise comparisons were performed with a Mann-Whitney Signed Ranks test ($n$ = 7-11 biological replicates; **** denotes $p < 0.00005$). (**B**) Invading cells (yellow, though absent in B) are killed when phages are introduced immediately after their arrival. Resident biofilm cells are shown in red. (**C**) Invading cells are not killed when phages are introduced 10 hr after their arrival.

The online version of this article includes the following source data and figure supplement(s) for figure 2:

**Source data 1.**

**Figure supplement 1.** Method for phage posttreatment experiments.

protection over this delay period. Given that phage protection is dependent on being embedded in curli polymers of the biofilm matrix (*Vidakovic et al., 2018*), it is possible that colonizing cells produce their own curli after an initial transition period following attachment to the resident biofilm. Alternatively, invading cells could co-opt matrix produced by the resident population. We tested these possibilities using a combination of experiments in which either the invading strain or the resident strain produced a 6xHis-labeled variant of CsgA, allowing us to localize and quantify curli production as a function of time and space inside biofilms via immunostaining. His-labeling of CsgA has been shown previously not to influence curli function in live biofilms (*Vidakovic et al., 2018*; *Serra et al., 2013b*). We cultivated resident biofilms and performed the invasion assay as above, but in this case we included an anti-His, AlexaFluor-647-conjugated antibody in the inflowing medium such that any curli produced in the biofilm became fluorescent and detectable by confocal microscopy (*Figure 3A*).

When invading cells harbored the His-labeled variant of *csgA*, we detected negligible anti-His fluorescence in the 10 hr following invasion of the biofilm exterior (*Figure 3—figure supplement 1*). It was not clear why colonizing cells do not more rapidly produce their own curli, despite occupying conditions presumably identical to those experienced by resident cells on the biofilm surface; we speculate that the progression of the curli expression program must be quite slow relative to the time scale of surface occupation of the invading strain in our experiments. This could be the case, for example, if curli production were dependent on cell-cell packing that emerges gradually over the course of biofilm growth. By contrast, when cells in the resident biofilm harbored the His-labeled variant of *csgA*, abundant anti-His staining was detectable in the upper surface of the resident biofilm (*Figure 3B–F*), including the surroundings of the colonizing cells (*Vidakovic et al., 2018*; *Serra et al., 2013a*; *Serra et al., 2013b*). These two observations indicate that invading cells do not gain phage protection in the 10 hr following colonization because they produce their own curli fibers, but rather because they coopt the curli fibers being produced by cells in the resident biofilm (*Figure 3B*). To support this interpretation, we used image analysis to quantify curli accumulation in the immediate neighborhood of resident and colonizing cell biomass following invasion; this analysis showed a steadily increasing amount of curli in the immediate neighborhood of colonizing cells over the course of 10 hr following their arrival to the outer surface of the resident biofilm. At the 10 hr

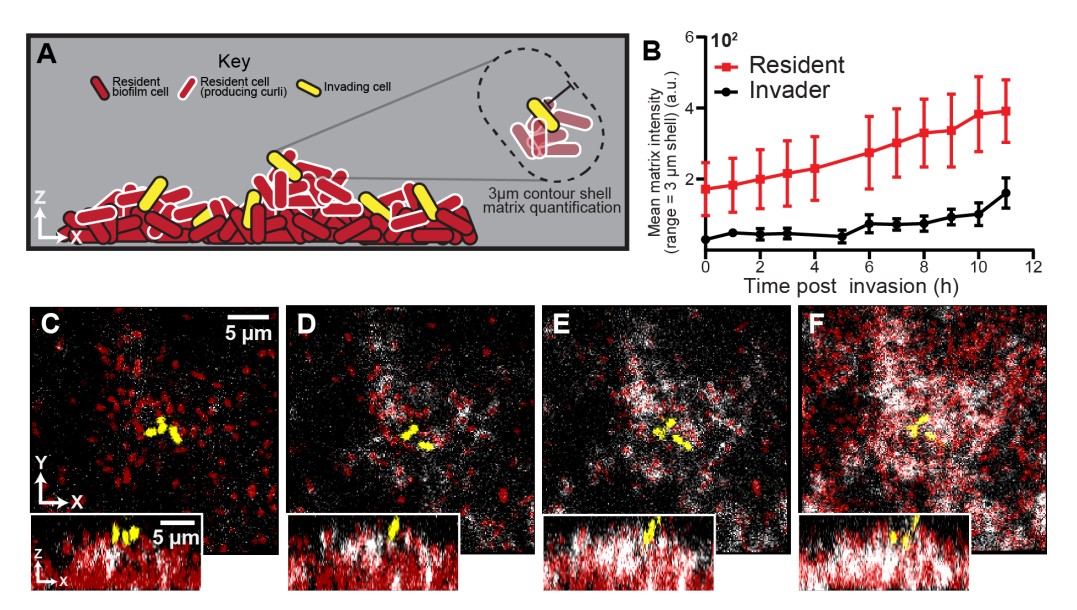

**Figure 3.** Spatial and temporal dynamics of curli fiber localization around resident and invading cells. (A) Illustration of matrix quantification method. Localized curli matrix was quantified by measuring CsgA-His immunofluorescence in 3 μm shells surrounding individual segmented cell volumes. (B) Quantification of matrix localization surrounding the resident and invading strain populations during an 11 hr time course after invading cells arrive (*n* = 3-4 biological replicates per time point, errors bars denote SEM). (C–F) Representative images displaying resident cells (red) producing labeled curli matrix (white); invading cells are shown in yellow. Images in (C-F) were acquired respectively at 1 , 5, 7, and 10 hr following the arrival of invading cells.

The online version of this article includes the following source data and figure supplement(s) for figure 3:

**Source data 1.**

**Figure supplement 1.** Invading cells do not produce curli matrix.

**Figure supplement 2.** Quantification of invasion success of *E. coli* lacking the CsgB baseplate required for curli polymerization, with phage exposure post-invasion.

**Figure supplement 2—source data 1.**

mark, invading cells were surrounded by the same amount of curli as the resident strain at the beginning of the invasion experiment.

We next asked if the invading strain directly or indirectly co-opts curli matrix material from the resident biofilm. Invading cells producing CsgB baseplates could potentially collect freely diffusing CsgA monomers being released by the resident biofilm population, accumulating curli matrix material and thus phage protection. This might not necessarily be the case, as our experiments above showed that the invading cells were not producing CsgA, whose corresponding gene is in the same operon as *csgB*; one would typically not expect the production of one without the other. Alternatively, without directly sequestering resident-produced CsgA to their cell surface, invading cells could become enveloped within curli-producing clusters of the resident biofilm population with enough curli in their surroundings to block phage diffusion. This would be an indirect way of exploiting the phage protection of the resident biofilm's curli layer.

To differentiate between these two possibilities, we repeated the post-invasion phage pulse experiments using a Δ*csgB* mutant that cannot produce its own CsgB base plate, and which therefore cannot nucleate CsgA polymerization on its outer surface. Here, as above (see *Figure 2*), few cells survived if phages arrived immediately after invading cells colonized the resident biofilm surface (*Figure 3—figure supplement 2*). However, despite lacking CsgB curli baseplates, invaders were still protected after a 10-hr delay prior to phage exposure in the system (*Figure 3—figure supplement 2*). This prompted the interpretation that invading cells do not have to be able to directly polymerize CsgA on their exterior to gain phage protection over the 10 hr after attaching to the biofilm surface. Rather this result suggests that the invading cells – despite remaining distinct physiologically

from the resident cells by virtue of not producing CsgA or CsgB themselves – *indirectly* coopt curli produced by the resident biofilm by becoming sufficiently enveloped amidst clusters of resident cells that their exposure to incoming phages is greatly reduced or eliminated.

## Conclusions

We have shown that matrix-embedded T7 phages can remain infectious on the curli-protected *E. coli* biofilm surface and kill newly arriving susceptible bacteria. In this sense, biofilm-dwelling microbes, by trapping phages on the biofilm periphery, can incidentally weaponize them against incoming phage-susceptible cells. On the other hand, we found that if invading *E. coli* cells attach to a resident biofilm and have sufficient time to become entangled in the curli matrix, they too gain protection from subsequent phage exposure. This protection is obtained by indirect exploitation of the resident biofilm's curli matrix: invading cells did not significantly use resident-produced curli monomers to polymerize curli on their own exterior, but rather became sufficiently enveloped in curli-producing groups of resident biofilm cells that they were no longer exposed to an incoming phage attack.

Our observations bear an interesting analogy to those of *Barr et al., 2013*, who found that phages trapped in host mucosal linings can kill incoming bacteria (see also *Barr et al., 2015*). We speculate on the basis of our results here that phage entrapment and their blocking effect against bacterial colonization is important not just in host associated mucosal environments but even more broadly to many biofilm contexts in which phage-trapping matrix material could potentially influence the pattern of community succession. We would not say that obligate lytic phages like T7 evolved to serve this function per se; their evolutionary interest is to infect host cells, replicate, and spread to new hosts. But the biofilm matrix almost certainly did evolve in part to protect the cells within from external threats, including exposure to phages; when phages are trapped in the matrix and remain viable, they can subsequently influence whether other incoming bacteria (depending on their phage susceptibility) colonize the resident biofilm surface.

As a demonstration of principle, we studied here the effect of matrix-embedded T7 phages on the colonization ability of cells isogenic to those in the resident *E. coli* biofilm, showing that biofilm colonization by susceptible cells was effectively eliminated by the presence of phages on the biofilm surface. This does not constitute a proof of generality that matrix-trapped phages always block colonization by susceptible bacteria, but we would think that this phenomenon is not unique to *E. coli*. We hope our report will prompt further tests for other bacterial and phage species; recent reports from other groups have suggested matrix-dependent protection against and potentially sequestration of incoming phages, but it remains to be seen whether phages remain active against other incoming cells in these cases (*Darch et al., 2017*; *Díaz-Pascual et al., 2019*; *Dunsing et al., 2019*). The centrally important and open question for future work prompted by our results is whether multi-species biofilm consortia trap multiple phage types of different strain and species specificities, and whether some or all of these trapped phages have the potential to kill off invading cells of their target host range. In this regard, the spatial ecology of biofilm-phage interaction may play a key role in the successional dynamics of polymicrobial biofilm communities.

## Materials and methods

### Strains

*E. coli* strains used in these experiments were all derived from *E. coli* AR3110 and are listed in *Table 1* below. Each strain used for imaging contained a codon-optimized fluorescent protein construct (*mKO-κ, mKate2* or *mTFP1*) under the control of a constitutive $P_{taq}$ promoter. Fluorescent protein constructs were inserted using the traditional lambda-red recombineering method by amplifying constructs with primer overhangs corresponding to the *attB* site (*Datsenko and Wanner, 2000*). iProof High-Fidelity DNA Polymerase (Bio-Rad, Hercules, CA, USA) was used to amplify insertion sequences for fluorescent protein expression constructs. The *E. coli* Δ*trxA* deletion strain was also constructed using lambda-red recombineering. The deletion construct was made by amplifying a kanamycin resistance marker flanked by FRT sites. Primer overhangs added upstream and downstream regions flanking start and stop codons of the *trxA* locus for replacement of the full reading

**Table 1.** Strains, plasmids, and oligos used in this study.

| Strain | Relevant markers/Genotype | Source |
|---|---|---|
| *E. coli* | | |
| CNE 689 | AR3110 wild type P$_{tac}$-*mKo-κ* inserted at *attB* site | This study |
| CNE 762 | AR3110 wild type P$_{tac}$-*mKate2* inserted at *attB* site | This study |
| CNE 760 | AR3110 wild type P$_{tac}$-*mTFP* inserted at *attB* site | This study |
| CNE 284 | AR3110 Δ*csgB*::scar, P$_{tac}$-*mruby* inserted at *attB* site | (19) |
| CNE 773 | AR3110 CsgA with C-terminal 6x His Tag, P$_{tac}$-*mKo-κ* inserted at *attB* site | This study, (19) |
| CNE 691 | AR3110 Δ*trxA*::scar, P$_{tac}$-*mKo-κ* inserted at *attB* site | This study |
| CNE 198 | AR3110 wild type | (19) |
| T7 Phage | | |
| CNX 06 | T7 with *sfgfp* under phi 10 promotor control | (19) |

| Plasmid | Origin, marker | Comments | Templates, primers | Source |
|---|---|---|---|---|
| pCN754 | pR6Kγ, Kan | Housing plasmid/ template for P$_{tac}$-*mTFP* insert | CNO 198, CNO 199, pCN 752, pCN 664 | This study |
| pCN755 | pR6Kγ, Kan | Housing plasmid/ template for P$_{tac}$-*mKate2* insert | CNO 198, CNO 199, pCN 753, pCN 664 | This study |
| pCN664 | pR6Kγ, Kan | Housing plasmid/ template for P$_{tac}$-*mKo-κ* insert | CNO 198 CNO 199 | This study |
| pNUT1336 | pR6Kγ,Kan | pUC housing IDT-synthesized double *mKO* optimized for *E. coli* AR3110 | | This study |
| pCN753 | pR6Kγ, Kan | pUC housing IDT-synthesized double *mTFP* optimized for *E. coli* AR3110 | | This study |
| pCN752 | pR6Kγ, Kan | pUC housing IDT-synthesized double *mKate2* optimized for *E. coli* AR3110 | | This study |

| Primer name | Sequence (designed using Snapgene) | Description |
|---|---|---|
| CNO 198 | ACAACTTTTTGTCTTTTTA CCTTCCCGTTTCGCTCAAGT TAGTATttgacaattaatcatcggctcg | Universal primer to amplify new *E. coli* Ptac_FP construct, with attB integration tail |
| CNO 199 | TCCGGGCTATGAAATAGAA AAATGAATCCGTTGAAGCC TGCTTTTcatgggaattagccatggtcc | Universal primer to amplify new *E. coli* Ptac_FP construct, with attB integration tail |
| CNO 138 | ACAACGAAACCAACACGCCAGGCTTA TTCCTGTGGAGTTAT ATgtgtaggctggagctgcttc | Forward primer to amplify pKD3 FRT-Cm-FRT with homology to trxA up flank |
| CNO 139 | GCGTCCAGTTTTTAGCG ACGGGGCACCCGAACATG AAATTCCCCcatatgaatatcctccttagt | Reverse primer to amplify pKD3 FRT-Cm-FRT with homology to trxA down flank |

*Table 1 continued on next page*

| CNO 146 | GAATGGGCGTACAGTTATGAAAC | Forward primer to check sequence of trxA deletion in *E. coli*, 200 bp upstream |
| --- | --- | --- |
| CNO 147 | TGCCTGGTCACAGGAGAGT | Reverse primer to check sequence of trxA deletion in *E. coli*, 200 bp downstrm |
| CNO 179 | GTGGATTGGGAACCGAGCA | Sequencing primer for pCN 664 |
| CNO 180 | GGAGATCCCAGACTACTTCAAAC | Sequencing primer for pCN 664 |
| CNO 181 | gtcaagaccgacctgtcc | Sequencing primer for pCN 664 |
| CNO 182 | ggacatagcgttggctacc | Sequencing primer for pCN 664 |
| CNO 183 | caccaatttcatattgctgtaagtg | Sequencing primer for pCN 664 |
| CNO 223 | TCTTTCACTTCCAGGTTAATGGTG | Sequencing primer for pCN 754 |
| CNO 224 | CGTTTGTGATTGAAGGCGAAG | Sequencing primer for pCN 754 |
| CNO 225 | GTGTATGAAAGCGCGGTGG | Sequencing primer for pCN 754 |
| CNO 226 | GGAACGTATGTACGTTCGTGAC | Sequencing primer for pCN 754 |
| CNO 229 | CATAATCCACCTCCTTTACTGGTC | Sequencing primer for pCN 755 |
| CNO 230 | AAACCGTATGAAGGCACCC | Sequencing primer for pCN 755 |
| CNO 231 | AAAGAAACCTATGTGGAACAGCAT | Sequencing primer for pCN 755 |
| CNO 232 | CGAATGGTCCGGTTATGC | Sequencing primer for pCN 755 |
| CNO 173 | TTTGGATCCTCT AAGCTTCATcctag | Forward primer for amplification of IDT 2x-mKO *E. coli* optimized (kde1336 template) |
| CNO 174 | ctccagcctacactttGAATT CtttTCTAGAAAGGAGCTCatg | Reverse primer for kde1336 template with overlap to FRT-Kan from pKD4 |
| CNO 175 | GAATTCaaagtgtaggctggagctgc | Forward primer for FRT-Kan amplification from pKD4 template with overlap to IDT 2x-mKO from pNUT1336 |
| CNO 176 | ggaagaaatagcgcatgggaattagccatggtcc | Reverse primer for FRT-Kan from pKD4, with overlap to pSC101 from kde970 |
| CNO 177 | ctaattcccatgcgctatttcttccagaattgc | Forward primer for pSC101 from kde970 with overlap to FRT_Kan from pKD4 |
| CNO 178 | aaaGGATCCattggtgagaatccaagcactag | Reverse primer for amplifying pSC01 from kde970, with BamHI site for ligating to upstream fragment from pNUT1336 |

*Table 1 continued on next page*

| CNO 189 | aaaGGATCCttgacaattaatc atcggctcgtataatgcctaggc CTAAGCTTCATcctaggGACAc | introduction of Ptac (no lacO) into new FP construct vector for *E. coli* |
| CNO 190 | GCTCGCGGTAATTTTTTCGG | for sequencing of Ptac inserted with CNO 189 |

frame with the Kan cassette. FRT recombinase was subsequently expressed in trans to remove the kanamycin resistance marker.

## Phage propagation and titer

T7 lytic phages were propagated and lysates collected in a manner adapted from *Bonilla et al., 2016*. Briefly, AR3110 wild type *E. coli* cultures were grown overnight in 5 mL lysogeny broth at 37° C at 250 rpm in a New Brunswick orbital shaking incubator. The host strain was then back diluted 1:20 into 100 mL lysogeny broth and allowed to grow to mid exponential phase (0.3–0.5 $OD_{600}$). At this time, T7 phages were spiked in from a frozen stock and $MgSO_4$ was added to a final concentration of 5 mM. The culture was placed back into the incubator for 3–4 hr, until the culture clarified. The entire volume was then vacuum filtered (0.22 µm filter Millipore Sigma). Phage titer was determined by traditional plaque assay (*Adams, 1959*). Briefly, host *E. coli* was grown overnight and subcultured as described above to achieve mid exponential phase (0.3–0.5 $OD_{600}$). Phage preparation was serially diluted by passing 10 µL into 990 µL for 100-fold dilutions. Top agar (0.5% agar, lysogeny broth) was melted and aliquoted into 3 mL volumes. Subsequently, 50 µL of a dilution was added to each sample along with $MgSO_4$ (5 mM). Molten top agar was then poured evenly onto lysogeny broth plates and placed at 37°C for 3 hr. Plates were removed and plaques were counted in order to calculate plaque forming units (PFU) per milliliter.

## Biofilm phage pretreatment invasion assay

We measured attachment and growth of exogenously added planktonic cells to curli protected biofilms with and without the prior addition of T7 phage. *E. coli* AR3110 expressing mKate2 was cultured in 5 mL lysogeny broth overnight at 37°C at 250 rpm in a New Brunswick orbital shaking incubator. *E. coli* AR3110 was used due to its strong biofilm formation ability relative to other K12 domesticated lab strains of *E. coli*, and the literature history of establishing the timing and components of matrix expression in this strain background (*Simmons et al., 2020*; *Vidakovic et al., 2018*; *Serra et al., 2013a*; *Serra et al., 2013b*; *Serra and Hengge, 2014*). Cultures were then pelleted and washed twice with 0.9% NaCl and standardized to $OD_{600}=0.2$ prior to inoculation into microfluidic devices. The inoculum was incubated for 1 hr at room temperature (approximately 22°C). Media syringes (1 mL BD plastic) were prepared by loading 1 mL of 1% tryptone broth (W/V) and attaching a 25-gauge needle. Tubing (#30 Cole palmer PTFE ID 0.3 mm) was then carefully attached to the needle and syringes were subsequently placed in Harvard Apparatus syringe pumps. After affixing inlet and outlet tubing to the microfluidic devices, a 40 s pulse at 40 µL/min was conducted to remove unattached cells, before standard flow regime (0.1 µL/min) was started. Biofilms were grown at room temperature for 62 hr at which time, tubing was swapped from clean media to either purified phages ($2x10^8$ PFU/mL) or clean media control. Flow was continued at 0.1 µL/min for 1 hr. After phage pretreatment, tubing was again removed and switched to syringes containing invading cells expressing mKO-κ for three hours at 0.1 µL/min. Invading cells were prepared prior to this step. Cells were grown overnight as previously described before in lysogeny broth. Invading cells were then sub-cultured 1:20 into 100 mL of 1% tryptone broth for 3 hr at 37°C at 250 rpm. Cells were then pelleted, concentrated, and standardized to $OD_{600}=6.0$ (~$5x10^9$ CFU/mL). Following the conclusion of the invasion, microfluidic chambers were allotted 10 hr of incubation at room temperature under standard flow conditions for growth and phage infection to occur. Three to five image fields (150 µm x 150 µm x 25 µm; length x width x height, slice interval 0.5 µm) were then acquired on a Zeiss 880 LSCM and image analysis performed using BiofilmQ software tool (*Hartmann et al., 2021*).

## Biofilm phage posttreatment invasion assay

Attachment and growth of invading *E. coli* were measured under two different regimes. Invasions of resident biofilms were conducted prior to phage application in this assay; however, timing of the phage application was varied. Resident biofilms were cultured in the same manner described above in the pretreatment invasion assay. At 62 hr of growth, clean media tubing was exchanged for invading cells (prepared in an identical manner as above) and allowed to flow for 3 hr at 0.1 μL/min. Following the conclusion of the invasion, chambers were separated into two groups corresponding to phage treatment regime. Half of the chambers were exposed to phage treatment ($2 \times 10^8$ PFU/mL at 0.1 μL/min for 1 hr) immediately following the invasion, while the other half was incubated for 10 hr at room temperature prior to the phage treatment. In both regimes, images were acquired in the manner described above. Imaging took place 10 hr following their respective phage treatments.

## Curli matrix localization assay

Curli matrix monomers, CsgA, were labeled with a 6X-His tag as previously published (*Vidakovic et al., 2018*). Curli fibers were detected via direct fluorescent immunostaining with an α−6X-His antibody (Invitrogen) conjugated to a fluorescent dye, Alexafluor647. Antibody was added to clean media at 0.1 mg/mL and flowed continuously throughout the course of the experiment. Biofilms were grown as described above, using *E. coli* expressing labeled curli.

## Confocal microscopy and image analysis

All imaging was performed using a Zeiss 880 line-scanning confocal microscope with either a 10x/0.4NA or a 40x/1.2NA water objective to minimize axial aberration effects. Representative images for each experiment (with *n* independent replicates as indicated in each figure legend) were taken at random locations throughout the corresponding microfluidic devices. The sfGFP fluorescent protein was excited using a 488 nm laser line, the mKO-κ fluorescent protein was excited using a 543 nm laser line, the mKate2 fluorescent protein was excited using a 594 nm laser line, and Alexafluor647 was excited using a 633 nm laser line. All image stacks were trimmed if necessary (e.g. if area outside of the microfluidic devices had been acquired in addition to the biofilm itself) using the native Zeiss Zen Blue software. All subsequent quantifications were performed using the BiofilmQ image analysis framework (*Hartmann et al., 2021*).

## Acknowledgements

The authors are grateful to members of the Nadell lab at Dartmouth for feedback on the project. MCB was supported by a GANN Fellowship from Dartmouth College and NIH grant P30-DK117469 to the Dartmouth Cystic Fibrosis Research Center. KD is supported by the European Research Council (StG-716734), the Deutsche Forschungsgemeinschaft (DR 982/5–1), and the Behrens-Weise-Foundation. CDN is supported by the Simons Foundation Award Number 826672, NSF grant MCB 1817342, NSF grant IOS 2017879, a Burke Award from Dartmouth College, a pilot award from the Cystic Fibrosis Foundation (STANTO15RO), NIH grant P30-DK117469, NIH grant 2R01AI081838 to PI Robert Cramer, NIH grant P20-GM113132 to the Dartmouth BioMT COBRE, and grant RGY0077/2020 from the Human Frontier Science Foundation with co-PI Alexandre Persat.

## Additional information

### Funding

| Funder | Grant reference number | Author |
| --- | --- | --- |
| Dartmouth College | GANN Fellowship | Matthew C Bond |
| National Institutes of Health | P30-DK117469 | Matthew C Bond<br>Carey D Nadell |
| European Research Council | StG-716734 | Knut Drescher |
| Deutsche Forschungsgemeinschaft | DR 982/5–1 | Knut Drescher |

| | | |
|---|---|---|
| Behrens-Weise-Foundation | | Knut Drescher |
| Simons Foundation | 826672 | Carey D Nadell |
| National Science Foundation | MCB 1817342 | Carey D Nadell |
| National Science Foundation | IOS 2017879 | Carey D Nadell |
| Dartmouth College | Burke Award | Carey D Nadell |
| Cystic Fibrosis Foundation | STANTO15RO | Carey D Nadell |
| National Institutes of Health | 2R01AI081838 | Carey D Nadell |
| National Institutes of Health | P20-GM113132 | Carey D Nadell |
| Human Frontier Science Program | RGY0077/2020 | Carey D Nadell |

The funders had no role in study design, data collection and interpretation, or the decision to submit the work for publication.

## Author contributions
Matthew C Bond, Conceptualization, Resources, Data curation, Formal analysis, Validation, Investigation, Visualization, Methodology, Writing - original draft, Writing - review and editing; Lucia Vidakovic, Praveen K Singh, Resources, Methodology; Knut Drescher, Resources, Writing - review and editing; Carey D Nadell, Conceptualization, Resources, Data curation, Formal analysis, Supervision, Funding acquisition, Validation, Investigation, Visualization, Methodology, Writing - original draft, Project administration, Writing - review and editing

## Author ORCIDs
Matthew C Bond ![ORCID] https://orcid.org/0000-0002-7188-9252
Lucia Vidakovic ![ORCID] http://orcid.org/0000-0002-5289-5163
Praveen K Singh ![ORCID] http://orcid.org/0000-0002-0254-7400
Knut Drescher ![ORCID] http://orcid.org/0000-0002-7340-2444
Carey D Nadell ![ORCID] https://orcid.org/0000-0003-1751-4895

## Decision letter and Author response
Decision letter https://doi.org/10.7554/eLife.65355.sa1
Author response https://doi.org/10.7554/eLife.65355.sa2

## Additional files

### Supplementary files
• Transparent reporting form

### Data availability
Raw data for the entire study has been provided in the source data files.

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
