## [Decision Letter]

**Acceptance summary:**

Biofilm matrix-dependent phage protection has been observed in a variety of bacteria. Using techniques including microfluidic culture, single-cell resolution confocal microscopy, and phage infection reporter, authors show here that phages are trapped in curli fibres of outer *E. coli* biofilm layer, and that these phages can attack newly-arriving susceptible cells.

**Decision letter after peer review:**

Thank you for submitting your article "Matrix-trapped viruses can protect bacterial biofilms from invasion by colonizing cells" for consideration by *eLife*. Your article has been reviewed by 3 peer reviewers, including Wenying Shou as the Reviewing Editor and Reviewer #1, and the evaluation has been overseen Gisela Storz as the Senior Editor.

Summary:

Biofilm matrix-dependent phage protection has been observed in a variety of bacteria. Using techniques including microfluidic culture, single-cell resolution confocal microscopy, and phage infection reporter, authors show here that phages are trapped in curli fibres of outer *E. coli* biofilm layer, and that these phages can attack newly-arriving susceptible cells.

Essential Revisions:

The reviewers are concerned about the generality of your findings. On the other hand, reviewers realize that perhaps, the strength of this paper is in quantitative methodology. Thus, you may choose to make it a Method/Resource paper (but you will need to state the novelty and the applicability of the Methods), or you can do more experiments to argue that the phenomenon is observed for other type(s) of phages.*Reviewer #1:*

Biofilm matrix-dependent phage protection has been observed in a variety of bacteria. Using techniques including microfluidic culture, single-cell resolution confocal microscopy, and phage infection reporter, authors show here that phages are trapped in curli fibres of outer *E. coli* biofilm layer, and that these phages can attack newly-arriving susceptible cells. Invading cells that arrived at the biofilm 10 hrs prior to phage arrival are protected by using the existing curli fibre as protection.

The experiments are nicely illustrated and well-done. The article is also well-written. I just have one question: What happens to the long-term fate of biofilm? Presumably, a fraction of cells can still grow in the biofilm and a fraction of cells will detach from the biofilm. How might the presence of phage affect these processes? Speculation is fine.*Reviewer #2:*

The motivation for this study is right on and important. In the real world, bacteria and bacteriophage (phage) live in physically structured habitats, where the bacteria exist as colonies or microcolonies are often embedded and stuck together in polymeric matrices known as biofilms. As the authors point out, most theoretical and experimental studies neglect this inconvenient reality. They use mass action models and their empirical analog, bacteria, and phage in well-agitated liquid culture. A full understanding of the population biology, ecology, and evolution of bacteria and phages require an understanding of how these populations intact in physically structured habitats.

This state-of-the-art study presents compelling evidence that the phage T7, which the authors have previously shown to be embedded in the matrices of biofilms are viable and capable of replicating on sensitive bacteria. It is not clear why these phages would not be viable and capable of replicating under these conditions. Is there evidence that suggests that phages are unable to replicate in biofilm populations of their host bacteria? If so, they should present this evidence.

This study considers only one form of physically structured bacterial populations, biofilms of *E. coli* on the surfaces of plastic tubes, and a single phage, T7. Does this result not obtain when the *E. coli* exist as colonies? Is this result unique to this rapid lysis phage? Will the same results obtain with other lytic phages, like T4? On surfaces and soft agar, the phage infection dynamics in these physically structured populations are different for these phages. T4 will be an excellent phage to examine whether rapid lysis is needed for this result; rapid lysis mutants of T4 can be obtained.

This study seems to have an agenda to provide evidence for, rather than broadly test, the hypothesis that lytic phages embedded in biofilms reduce the likelihood of those biofilms being colonized by other bacteria. The demonstration that matrix embedded phage T7 can reduce the likelihood of *E. coli* colonizing an established biofilm under the limited conditions of this experiment is of little general interest. Biofilms are commonly composed of multiple species of bacteria. The host range of phages is usually limited within a species and among strains of those species. The embedded phages of one species or strain of bacteria will not prevent colonization by other bacterial species.

More consideration should be given to the population and evolutionary dynamics of the phage in biofilm populations of bacteria. Do the phages in these *E. coli* biofilms continue to replicate and maintain their populations? What about resistance? As is commonly seen in liquid cultures of bacteria and lytic phages, resistance evolves, and the dominant population of bacteria is not susceptible to the phage. It would be of some interest and add generality to this study to know if the *E. coli* in the biofilms are sensitive or resistant to T7, and how that affects their premise that these embedded phages prevent colonization biofilms.

Biofilms are often associated with the pathogenesis of bacteria and resistance to antibiotics. It would be of some interest to know what the results presented in this study relate to these practical elements of biofilms' biology.

Although well-designed and well-executed, the inferences drawn from his study of *E. coli* and T7 in biofilms are too limited to support the publication of this manuscript in *eLife*.

Specific Comments and Suggestions:

– The abstract does a fine job of summarizing the main points of the paper.

– Line 44: How do they differ? Expand!

– Line 52: The articles cited do not provide compelling support for the statement that biofilms shape bacteria and phage evolution. This statement, which seems fundamental to this study's motivation, requires expansion and support from the literature.

– In general, the introduction provides the necessary background to understand the paper for microbiologists. On the other hand, the questions addressed in this study are not presented in a manner that would be of interest to the broad audience *eLife* aspires to.

– Line 80: The authors do not present the rationale behind the 62 hour-growth or biofilm confluent growth.

– Line 89: The densities of bacteria and phage in these experiments seem high. They should justify the choice of densities and multiplicities of infection. It will be of some interest if the same qualitative results obtain with seemingly more relativistic lower densities of bacteria and phage and different starting conditions. Of particular interest in this regard is Figure 1

– Line 99: There is no consideration of the dynamics of phage infection, particularly the latent period. Could the infecting population of bacteria replicate? Are they all killed by the embedded phage?

– Curiously, they emphasize their elegant and cool technology but don't the rigorous experiments and controls necessary to test their hypothesis that embedded phages prevent the colonization of biofilm populations of bacteria.

– Line 116: The authors need to provide the rationale behind selecting the pulse times (10 – hours). Is this period supported by previous findings that are not presented?

–

– Line 168: The authors present a conclusion at the end of this paragraph that could be considered an example of wishful thinking. The result presented seems to be inconsistent with their hypothesis. Nevertheless, they present these results as indirect evidence in support of their hypothesis.

– Line 173: From an evolutionary perspective, it would be difficult to develop a mechanism by which bacteria domesticate phage for their protection. They should consider the selective pressures responsible for the phage protecting the biofilm populations of bacteria from invasion by other bacteria. Is it a coincidental or evolved phenomenon?

– The authors implicitly assume the phage-bacteria interaction is that of a predator (parasitoid, really) and its host, negatively impacting the bacterial population. They may want to point out; their results suggest that in biofilms, the association between phage and bacteria is more that of a symbiont and its host. If this is correct, they should point this out.

– The authors should clearly state that their observations, albeit well-supported, may be specific for the model system they used and for strictly virulent phage infections.

– To a large extent, this report can be described as a methods paper, a proof in principle that the elegant methods they present can be used to address some of the interactions between phage and bacteria in biofilms. We suggest the authors focus the paper on the utility and the novelty of the technique used, rather than the role of virulent phage infection for protecting bacteria in biofilms from colonization by other bacteria.

– Figure 1 would be clearer if it was divided into three figures. The diagram (cartoon) could be presented as supplemental material since it is not required to understand the results presented in the manuscript's body. Initially, we found it difficult to understand the results presented in I, J and K are about.

– Figure 2 provides the information needed to understand this control experiment. But as suggested earlier, the logic behind the 10 hours of the pulse has to be provided.

– Figure 3 is hard to understand. The cartoon in Figure 3A does not provide a clear explanation of the experiment.

– The three main figures of the paper illustrate the method rather than provide evidence in support of their hypothesis. As elegant as the methods presented are, we don't consider this technical contribution appropriate for publication in *eLife*.

– In Supplemental Table S1. The authors should explain why some nucleotides are presented in upper case and others in lower case.

– The software used to design the primers should be made available.

– Figure S4. They can improve the resolution of this figure.*Reviewer #3:*

This short report builds on work by Vidakovic et al. (1) that had demonstrated protection of *E. coli* cells growing in biofilms against killing by bacteriophage T7 through sequestration of T7 phage particles by curli, a protein component of the *E. coli* biofilm matrix. In the work presented here, Bond et al. provide evidence that these entrapped T7 phage particles are still infectious and can kill invading *E. coli* cells that would otherwise be able to integrate into the biofilm. They further show that cells that invade the biofilm before T7 is introduced can integrate into the biofilm and acquire protection from killing by the phage. Intriguingly, this protection appears to be granted by curli produced by the resident cell population rather than curli produced by the invading cells themselves.

The data are convincing and clearly presented, and the authors' arguments are well supported. The findings are a significant experimental advance in the largely theoretical field of phage-host dynamics in structured communities.

There are a few opportunities for improvement in clarity or content:

A) Although the localization of matrix-entrapped T7 particles can be inferred by the patterns and cell death and curli staining shown here, it is never explicitly shown. The foundational paper for this report (1) did show by microscopy that fluorescently labeled T7 accumulates at the biofilm periphery in a curli- and flagellin-dependent manner and that fluorescently labeled T7 colocalizes with fluorescently labeled curli. Therefore, while not strictly necessary to show colocalization of T7 and curli in these experiments, it might strengthen the work to explicitly refer readers to the previous work.

B) What are the precedents for His-tagged curli monomers (with and without fluorescently antibody labeling) polymerizing and performing their normal function in *E. coli* biofilms? It's possible that matrix containing a significant amount of tagged protein and/or antibody might have altered functional properties compared to wild-type matrix.

C) What is the interpretation of the difference between the distributions of invading cell cluster sizes shown in Figures 1J and 1K? It's not necessarily an intuitive result. One might imagine that the phage burst resulting from lysis of an invading cell would lead to a locally high phage density and a high chance of the entire invading cell cluster being eliminated rather than some small number of cells surviving-something like a microscopic analog of clear plaques on a plate.

D) It's curious that invading cells, being in locally near-identical environments to their neighboring curli-producing resident cells, do not begin producing curli within 10 h of colonization. Speculation on this phenomenon would be welcome. If invading cells do eventually begin producing curli, what toggles the switch? If not, how do resident and invader cells "know" that they belong to different populations despite being congenic?

E) Image acquisition and analysis could be described in more detail. As is, it's unclear how many fields of view are examined per experimental replicate and how the fields of view were selected (for instance, in invasion experiments are the fields selected randomly for biovolume calculation or are fields showing invasion events specifically selected?).

Reference:

1. Vidakovic L, Singh PK, Hartmann R, Nadell CD, Drescher K. 2018. Dynamic biofilm architecture confers individual and collective mechanisms of viral protection. Nat Microbiol 3:26-31. doi:10.1038/s41564-017-0050-1

---

## [Author Response]

Reviewer #1:Biofilm matrix-dependent phage protection has been observed in a variety of bacteria. Using techniques including microfluidic culture, single-cell resolution confocal microscopy, and phage infection reporter, authors show here that phages are trapped in curli fibres of outer *E. coli* biofilm layer, and that these phages can attack newly-arriving susceptible cells. Invading cells that arrived at the biofilm 10 hrs prior to phage arrival are protected by using the existing curli fibre as protection.The experiments are nicely illustrated and well-done. The article is also well-written. I just have one question: What happens to the long-term fate of biofilm? Presumably, a fraction of cells can still grow in the biofilm and a fraction of cells will detach from the biofilm. How might the presence of phage affect these processes? Speculation is fine.

We would like to thank the referee for their positive evaluation of our work, and their inquiry regarding the downstream consequences of phage trapping for biofilm-dwelling cells. This is an excellent question and one which we hope to pursue in the future experimentally. For the time being, we can say that with sufficient time, biofilms of *E. coli* grow large enough that they eventually slough off of the surface to which they are attached. As our work shows that matrix-trapped phages remain infectious, the phage population presumably disperses with them and may infect newly planktonic cells. A key question is whether the resulting phage propagation is sufficient to halt further surface colonization by surviving bacteria and another round of biofilm growth. We have added a note in the text to this end, and we look forward to reporting on this question in future work (Lines 132-135).

Reviewer #2:The motivation for this study is right on and important. In the real world, bacteria and bacteriophage (phage) live in physically structured habitats, where the bacteria exist as colonies or microcolonies are often embedded and stuck together in polymeric matrices known as biofilms. As the authors point out, most theoretical and experimental studies neglect this inconvenient reality. They use mass action models and their empirical analog, bacteria, and phage in well-agitated liquid culture. A full understanding of the population biology, ecology, and evolution of bacteria and phages require an understanding of how these populations intact in physically structured habitats.

We very much appreciate the referee’s agreement on the timeliness of the research questions pursued in our paper. We are very much of the same mind that phage-host interactions in nature can be completely different in the biofilm context. The referee brings up a number of important and constructive criticisms below, and we have endeavored at all junctures to address them. Some would require more experimental work than we currently have the ability to perform, so we are hopeful that a combination of new experiments and expanded discussion will be sufficient.

This state-of-the-art study presents compelling evidence that the phage T7, which the authors have previously shown to be embedded in the matrices of biofilms are viable and capable of replicating on sensitive bacteria. It is not clear why these phages would not be viable and capable of replicating under these conditions. Is there evidence that suggests that phages are unable to replicate in biofilm populations of their host bacteria? If so, they should present this evidence.

We appreciate the referee’s comments here. The rationale for questioning whether curli-trapped phages might not be active against cells is that they either degrade after being embedded in the matrix, or that they become embedded in orientations that prevent them from being able to readily infect cells that are arriving on the biofilm surface. This possibility had not been tested previously to our knowledge. For biofilms that are not protected by matrix (including *E. coli* biofilms before they begin producing curli fibers) phages are able to diffuse among and infect biofilm-dwelling bacteria. This was shown previously by the Vidakovic et al. paper to which the referee refers in the comment above. To clarify this point, we have added new text to the introduction in support of the rationale for the paper (Lines 59-62).

This study considers only one form of physically structured bacterial populations, biofilms of E. coli on the surfaces of plastic tubes, and a single phage, T7. Does this result not obtain when the *E. coli* exist as colonies? Is this result unique to this rapid lysis phage? Will the same results obtain with other lytic phages, like T4? On surfaces and soft agar, the phage infection dynamics in these physically structured populations are different for these phages. T4 will be an excellent phage to examine whether rapid lysis is needed for this result; rapid lysis mutants of T4 can be obtained.

These are all excellent suggestions, and we appreciate the insight on the part of the referee. The spatiotemporal matrix composition of colonies grown on agar and biofilms grown in flow chambers may differ substantially, although Regine Hengge’s work and the Vidakovic et al. paper also show some surprising similarities in matrix localization despite the different directions from which nutrients enter the cell groups in the different assays. While it would certainly be interesting to determine if our results hold in agar colony models, the level of single-cell level resolution fluorescence imaging required for our work is generally not possible for agar colonies. A main goal of the present paper, which demonstrates that matrix-trapped phages kill off invading cells, is to motivate further work on this question by others in the biofilm and phage fields using different growth conditions, host bacteria, and phages. We would respond similarly for the question of whether the same results hold for phage T4. We are not able to make this assessment at present due to lab restrictions, but testing the generality of these results is an important avenue for further work. We added new emphasis in the conclusion section of the paper to make it clear that our work established a proof of principle, but not a proof of generality across all environments, all bacteria, and all phages, and that we hope other researchers in the field will join us to address this question in the future (Lines 231-236).

This study seems to have an agenda to provide evidence for, rather than broadly test, the hypothesis that lytic phages embedded in biofilms reduce the likelihood of those biofilms being colonized by other bacteria. The demonstration that matrix embedded phage T7 can reduce the likelihood of *E. coli* colonizing an established biofilm under the limited conditions of this experiment is of little general interest. Biofilms are commonly composed of multiple species of bacteria. The host range of phages is usually limited within a species and among strains of those species. The embedded phages of one species or strain of bacteria will not prevent colonization by other bacterial species.

While we understand the skepticism on the part of this referee, we do not agree that our results will be of little general interest. First, the microfluidic flow conditions used in our experiments are an excellent model system that reflects many natural settings in which biofilms grow on a solid-liquid interface. Though we only examine a single-species and single-phage condition, this is the first result of its kind and meant as a demonstration of principle, not proof of universality, to spark interest in the question of how trapped phages influence bacterial colonization of resident. We agree that since phages are species and strain-specific, they will not kill all invading bacteria; in fact, we demonstrate this directly using a T7-resistant mutant of invading *E. coli* in the paper. Our results prompt the question of whether natural biofilms might trap phages of many different origins and then become resistant to colonization by the species/strains to which those trapped phages are virulent. An important future step is to determine whether this is the case, as we originally highlighted in the conclusion section of the paper. With this referee comment in mind, though, we understand we did not emphasize the caveats of our paper in sufficient depth, and we have expanded this section of the conclusion (to the extent possible given space constraints for *eLife* short reports) to make it clear what questions are left open by our results. We hope this modification to the conclusion will be more effective in prompting the readership to include similar assays in their work (Lines 236-244).

More consideration should be given to the population and evolutionary dynamics of the phage in biofilm populations of bacteria. Do the phages in these E. coli biofilms continue to replicate and maintain their populations? What about resistance? As is commonly seen in liquid cultures of bacteria and lytic phages, resistance evolves, and the dominant population of bacteria is not susceptible to the phage. It would be of some interest and add generality to this study to know if the *E. coli* in the biofilms are sensitive or resistant to T7, and how that affects their premise that these embedded phages prevent colonization biofilms.Biofilms are often associated with the pathogenesis of bacteria and resistance to antibiotics. It would be of some interest to know what the results presented in this study relate to these practical elements of biofilms' biology.

These are also appreciated criticisms from the referee, and we can address the majority based on our previously published work. The phages trapped in the curli matrix might be able to replicate to a degree, but not to the extent that they any major impact on net positive growth of the biofilm. This was shown by Vidakovic et al. (2018). But overall, the phages are trapped in the curli matrix of mature *E. coli* biofilms, and neither proliferate nor degrade. Our central question in this paper was whether these phages can in principle infect and replicate and newly arriving, non-curli-protected cells arrive, and we find that the answer is yes.

We have shown in a previous publication that in a naïve resident biofilm typical of those used in our experiments, there are about 4x10^6^ resident *E. coli*, of which ~15 cells are de novo resistant mutants. In all likelihood these mutants are buried in the biofilm in random locations and remain unexposed to phages blocked on the biofilm periphery. Among cells in the invading population, clearly few or none are resistant mutants because they are all killed by phages trapped on the resident biofilm exterior. The population sizes we are working with are orders of magnitude smaller than those in well-mixed experiments in test tubes – because the number of de novo resistant *E. coli* cells is small relative to the whole population, and phage-host interactions are spatially constrained, phage resistance evolution is very unlikely to make any difference to the outcome of our experiments. We have added new text stating this point and appreciate the note from the reviewer to make sure there is no confusion among readers (Lines 85-87).

Although well-designed and well-executed, the inferences drawn from his study of *E. coli* and T7 in biofilms are too limited to support the publication of this manuscript in eLife.Specific Comments and Suggestions:– The abstract does a fine job of summarizing the main points of the paper.– Line 44: How do they differ? Expand!

We expand on this point in the following paragraph of the introduction.

– Line 52: The articles cited do not provide compelling support for the statement that biofilms shape bacteria and phage evolution. This statement, which seems fundamental to this study's motivation, requires expansion and support from the literature.

We have cited all the papers that we could find directly examining the impact of phage on host evolution in the biofilm context, including our own recently published work to this effect. We’re afraid that without further clarification from the referee on what was missing here, we are not able to expand further. We are also working with the limited word space of the short report format of *eLife*, and so not every introductory point can be expanded upon in full.

– In general, the introduction provides the necessary background to understand the paper for microbiologists. On the other hand, the questions addressed in this study are not presented in a manner that would be of interest to the broad audience eLife aspires to.

While we appreciate the reviewer’s concern about general interest, we do not agree that this is the case. This has been evident in numerous departmental talks and a biofilm conference at which this material has been recently presented and exceptionally well received by broad audiences. We are quite confident in the general interest of this work and therefore feel it is an excellent fit to *eLife.* We hope that after the clarifications noted above and below, that the referee will have change of heart about this.

– Line 80: The authors do not present the rationale behind the 62 hour-growth or biofilm confluent growth.

This point from the referee is much appreciated, and to support our addition to the manuscript we have performed additional experiments showing that it is at the ~62 h mark that resident biofilms of *E. coli* develop enough of a curli matrix layer to effectively block phage diffusion and trap phages in place on the biofilm exterior. We have added text and a new supplemental figure in light of this comment (Lines 91-93).

– Line 89: The densities of bacteria and phage in these experiments seem high. They should justify the choice of densities and multiplicities of infection. It will be of some interest if the same qualitative results obtain with seemingly more relativistic lower densities of bacteria and phage and different starting conditions. Of particular interest in this regard is Figure 1.

We are afraid that this is not possible to do. In order to obtain replicable *E. coli* biofilm conditions, we must start experiments with the surface density of cells that is shown in the first submission. As noted above, the subsequent growth period for 62 hours is needed for sufficient curli production to trap phages after exposure. Presumably, the effect of phages killing incoming susceptible cells might be weakened with very few phages trapped in the matrix, i.e. lower MOI. In principle this could be tested in the future, but given the difficulty of these experiments and current limitations on lab work, this is not feasible at the present time.

– Line 99: There is no consideration of the dynamics of phage infection, particularly the latent period. Could the infecting population of bacteria replicate? Are they all killed by the embedded phage?

To our knowledge, when invading bacteria were added to biofilms with phages embedded in the curli matrix, the invading bacteria were nearly all killed. The only invading cells we could find were expressing the phage infection reporter, indicating that they were in the middle of the phage latent period. This point is highlighted in the main text.

– Curiously, they emphasize their elegant and cool technology but don't the rigorous experiments and controls necessary to test their hypothesis that embedded phages prevent the colonization of biofilm populations of bacteria.

This criticism is not possible to address because it does not specify which experiments are missing. We feel that all of the needed controls are present in Figure 1, to establish the core point that matrix-embedded phages can prevent the colonization of resident biofilms by phage-susceptible (but not phage-resistant) bacteria. The clarity of this result seems to be supported by the comments from the other referees as well.

– Line 116: The authors need to provide the rationale behind selecting the pulse times (10 – hours). Is this period supported by previous findings that are not presented?

This was the duration of time over which we observed the biofilm population dynamics to come to equilibrium after phage pulse. This is now noted in the text (Lines 142-143).

–– Line 168: The authors present a conclusion at the end of this paragraph that could be considered an example of wishful thinking. The result presented seems to be inconsistent with their hypothesis. Nevertheless, they present these results as indirect evidence in support of their hypothesis.

We appreciate the skepticism on the part of the referee, but here the referee does not specify exactly what they consider to be wishful thinking in the interpretation of these results. Furthermore, the hypothesis at the end of this paragraph is explicitly presented as interpretation.

– Line 173: From an evolutionary perspective, it would be difficult to develop a mechanism by which bacteria domesticate phage for their protection. They should consider the selective pressures responsible for the phage protecting the biofilm populations of bacteria from invasion by other bacteria. Is it a coincidental or evolved phenomenon?

This point is much appreciated and one which we did not communicate clearly enough in the original text. The matrix evolved without questions to address many environmental challenges, including resisting fluid shear, controlling cell spatial localization, and protection from external threats including phages and antibiotics. The fact that trapped phages can act as a defense against invasion by competitors may well be incidental, but nevertheless could contribute to the adaptive value of matrix production. We have changed our wording in the conclusion to reflect this point (Lines 226-230). We have also changed our title to be more descriptive of the main result to help avoid confusion.

– The authors implicitly assume the phage-bacteria interaction is that of a predator (parasitoid, really) and its host, negatively impacting the bacterial population. They may want to point out; their results suggest that in biofilms, the association between phage and bacteria is more that of a symbiont and its host. If this is correct, they should point this out.

We have adjusted the text in light of this comment from the referee (Line 226).

– The authors should clearly state that their observations, albeit well-supported, may be specific for the model system they used and for strictly virulent phage infections.

We have added text to the conclusion section to better emphasize this point (Lines 231-236).

– To a large extent, this report can be described as a methods paper, a proof in principle that the elegant methods they present can be used to address some of the interactions between phage and bacteria in biofilms. We suggest the authors focus the paper on the utility and the novelty of the technique used, rather than the role of virulent phage infection for protecting bacteria in biofilms from colonization by other bacteria.

We respectfully do not agree with this point from the referee. Though we are grateful for the referee’s appreciation of our technical approach, we do still feel that the highlight of this work is the ecological/biological result that matrix-trapped phages can in principle block the colonization of the biofilm by other bacteria that are susceptible to the trapped phages. Though this observation needs to be tested for other species and environments, this core result has not been reported previously and will, we are confident, prompt others in the field to think in a new way about biofilm-phage interactions. This hope has been borne out in numerous presentations to biofilm-specific and broad-audience lecture settings, so we do not think we are being unrealistic in this regard.

– Figure 1 would be clearer if it was divided into three figures. The diagram (cartoon) could be presented as supplemental material since it is not required to understand the results presented in the manuscript's body. Initially, we found it difficult to understand the results presented in I, J and K are about.

We’re afraid this is not possible given the figure constraints of this publication format for *eLife*. We are also confident that all of the information in Figure 1 is best presented in one package for the sake of rigor and completeness in the experiments. We have added a new paragraph to clarify the interpretation of Figure 1J,K (Lines 115-124).

– Figure 2 provides the information needed to understand this control experiment. But as suggested earlier, the logic behind the 10 hours of the pulse has to be provided.

This has been addressed in response to the referee’s comment above.

– Figure 3 is hard to understand. The cartoon in Figure 3A does not provide a clear explanation of the experiment.

We’re afraid that without further specification as to what is unclear, we are not able to address this comment from the referee.

– The three main figures of the paper illustrate the method rather than provide evidence in support of their hypothesis. As elegant as the methods presented are, we don't consider this technical contribution appropriate for publication in eLife.

We respectfully disagree with this assessment, which is also not consistent with the feedback from the other two referees. The referee states here that the evidence presented does not justify the conclusions, but she/he does not specify in detail why they feel this is the case. Without more specific feedback, this criticism is not possible to address, nor is it consistent with the sentiments of the other referees on this submission.

– In Supplemental Table S1. The authors should explain why some nucleotides are presented in upper case and others in lower case.

This has been corrected.

– The software used to design the primers should be made available.

This has been corrected.

– Figure S4. They can improve the resolution of this figure.

This figure is at the limit of the resolution permitted by the confocal microscopy equipment we have access to.

Reviewer #3:This short report builds on work by Vidakovic et al. (1) that had demonstrated protection of *E. coli* cells growing in biofilms against killing by bacteriophage T7 through sequestration of T7 phage particles by curli, a protein component of the *E. coli* biofilm matrix. In the work presented here, Bond et al. provide evidence that these entrapped T7 phage particles are still infectious and can kill invading *E. coli* cells that would otherwise be able to integrate into the biofilm. They further show that cells that invade the biofilm before T7 is introduced can integrate into the biofilm and acquire protection from killing by the phage. Intriguingly, this protection appears to be granted by curli produced by the resident cell population rather than curli produced by the invading cells themselves.The data are convincing and clearly presented, and the authors' arguments are well supported. The findings are a significant experimental advance in the largely theoretical field of phage-host dynamics in structured communities.

We are delighted for the referee’s overall positive impression of our work, and we appreciate both the close reading of the paper and suggestions for improvement below.

There are a few opportunities for improvement in clarity or content:A) Although the localization of matrix-entrapped T7 particles can be inferred by the patterns and cell death and curli staining shown here, it is never explicitly shown. The foundational paper for this report (1) did show by microscopy that fluorescently labeled T7 accumulates at the biofilm periphery in a curli- and flagellin-dependent manner and that fluorescently labeled T7 colocalizes with fluorescently labeled curli. Therefore, while not strictly necessary to show colocalization of T7 and curli in these experiments, it might strengthen the work to explicitly refer readers to the previous work.

This comment is much appreciated, and we now make a clearer reference to the Vidakovic et al. (2018) paper with respect to the observation of matrix-trapped phages on the periphery of curli-expressing biofilms (Lines 52-54).

B) What are the precedents for His-tagged curli monomers (with and without fluorescently antibody labeling) polymerizing and performing their normal function in *E. coli* biofilms? It's possible that matrix containing a significant amount of tagged protein and/or antibody might have altered functional properties compared to wild-type matrix.

This is a helpful note as we did not mention the precedent for localizing curli via immunostaining of His-tagged csgA. This method has been used in several previous publications studying *E. coli* biofilm architecture by the Hengge and Drescher groups, with no indication of interference with normal matrix function (i.e. no major differences from WT biofilm morphology, shear resistance, etc.). We now make a more careful note of this in the main text (Lines 162-163).

C) What is the interpretation of the difference between the distributions of invading cell cluster sizes shown in Figures 1J and 1K? It's not necessarily an intuitive result. One might imagine that the phage burst resulting from lysis of an invading cell would lead to a locally high phage density and a high chance of the entire invading cell cluster being eliminated rather than some small number of cells surviving-something like a microscopic analog of clear plaques on a plate.

Thank you for pointing out that this portion of the figure needs some additional explanation in the text. Figure 1J and K depict the change in cluster size distribution of invading cells in the absence (J) and presence (K) of curli-trapped phages on the resident biofilm’s surface. They show that when phages are absent when invading cells arrive, the invading strain is able to establish cell clusters over time that most likely result from several rounds of division by one or a few cells that attach to a given location on the resident biofilm surface. When phages are present, on the other hand, the majority of invading cells are killed, and so there are almost no changes for larger cell clusters to develop. We have added new text to the Results section to make sure this interpretation is explained clearly (Lines 115-124).

D) It's curious that invading cells, being in locally near-identical environments to their neighboring curli-producing resident cells, do not begin producing curli within 10 h of colonization. Speculation on this phenomenon would be welcome. If invading cells do eventually begin producing curli, what toggles the switch? If not, how do resident and invader cells "know" that they belong to different populations despite being congenic?

On the basis of our His-labeling experiments, we are confident that the invading *E. coli* strain does not begin producing their own curli on the 10 h time scale before phages are introduced, but from a mechanistic perspective we are not entirely sure as yet why this is the case. From our experience, cells colonizing surface from the planktonic phase take at least 2 days before they begin to produce curli. Though the curli regulon has been explored in detail by other groups, it is not fully clear what mechanisms control this timing. We now add direction discussion of this point (Lines 170-175). We can speculate that invading cells, which do not appear to be expressing the *csg* operon and therefore most likely do not have a CsgB baseplate on their exterior, cannot directly coopt the CsgA produced by curli-producing resident biofilm cells, and that this controls the separation of the two populations despite the fact that they are congenic. As we show, though, the invading strain does appear to be folded into the growing front of the resident biofilm, despite the fact that the invading strain does not appear to be activating the *csg* operon. Additional commentary on this point is now provided at the end of this results subsection (Lines 206-210).

E) Image acquisition and analysis could be described in more detail. As is, it's unclear how many fields of view are examined per experimental replicate and how the fields of view were selected (for instance, in invasion experiments are the fields selected randomly for biovolume calculation or are fields showing invasion events specifically selected?).

We appreciate this input from the referee and have updated the methods section to make it clear that image locations were selected randomly, and to highlight sample sizes as directly as possible for each quantification.

Reference:1. Vidakovic L, Singh PK, Hartmann R, Nadell CD, Drescher K. 2018. Dynamic biofilm architecture confers individual and collective mechanisms of viral protection. Nat Microbiol 3:26-31. doi:10.1038/s41564-017-0050-1The basic story reminds me of the study (now 10 years old, maybe) out of Forest Rohwer's group showing that phage bind the mucous surrounding corals. The authors suggested that the phage provided a kind of defense system for the coral against bad bacteria. My thought was that phages sticking in the mucous probably included those that attacked good bacteria; it would be wishful thinking to suppose that phages were evolving to benefit the coral. The authors never mentioned this obvious point.In this case (from the abstract), as with Rowher's group, there is an implied purpose to the defense rather than more benign explanations. I don't like that, as it's almost as if they are trying to create a story. (The text may be more reasonable.)

We appreciate this feedback from the 4^th^ informal referee. We are also familiar with the Rohwer study, and in fact this was one of the main motivations for this paper. The Rohwer study and our results’ connection to it was a central part of the original submission’s Discussion section, and remains so. As to the interpretation that phages evolved to help the biofilms in which they become trapped, we never held this interpretation and agree with the referee that the fact that phages may help biofilms repel invading cells would be a coincidental benefit to the fact that the matrix is trapping the phages and preventing them from harming the biofilm-resident cells in the first place; this latter function is what we would presume to be the primary evolutionary explanation for the biofilm matrix- *i.e.* protecting the biofilm from threats, including phages. The fact that the trapped phages may help the biofilm-resident cells, even if a coincidental benefit, is nevertheless a potentially important ecological phenomenon. This is the main message of the paper. In response to this note and that of referee #2 above, we have modified the title of the paper to be more descriptive, and we have modified the text of the conclusion so as not to give the impression that we are arguing phages have evolved to help biofilms (Lines 228-233).